# Risk Assessment of Caffeine and Epigallocatechin Gallate in Coffee Leaf Tea

**DOI:** 10.3390/foods11030263

**Published:** 2022-01-19

**Authors:** Nadine Tritsch, Marc C. Steger, Valerie Segatz, Patrik Blumenthal, Marina Rigling, Steffen Schwarz, Yanyan Zhang, Heike Franke, Dirk W. Lachenmeier

**Affiliations:** 1Postgraduate Study of Toxicology and Environmental Toxicology, Rudolf-Boehm-Institut für Pharmakologie und Toxikologie, Universität Leipzig, Härtelstraße 16-18, 04107 Leipzig, Germany; nt60rawo@studserv.uni-leipzig.de (N.T.); Heike.Franke@medizin.uni-leipzig.de (H.F.); 2Chemisches und Veterinäruntersuchungsamt (CVUA) Karlsruhe, Weissenburger Strasse 3, 76187 Karlsruhe, Germany; Valerie.Segatz@stud.hs-coburg.de; 3Coffee Consulate, Hans-Thoma-Strasse 20, 68163 Mannheim, Germany; marcsteger2@googlemail.com (M.C.S.); patrik.blumenthal@live.de (P.B.); schwarz@coffee-consulate.com (S.S.); 4Department of Flavor Chemistry, University of Hohenheim, Fruwirthstrasse 12, 70599 Stuttgart, Germany; marina.rigling@uni-hohenheim.de (M.R.); yanyan.zhang@uni-hohenheim.de (Y.Z.); 5Hochschule für Angewandte Wissenschaften Coburg, Friedrich-Streib-Strasse 2, 96450 Coburg, Germany

**Keywords:** coffee leaf tea, novel food, coffee by-products, *Coffea arabica*, risk assessment, caffeine, epigallocatechin gallate

## Abstract

Coffee leaf tea is prepared as an infusion of dried leaves of *Coffea* spp. in hot water. It is a traditional beverage in some coffee-producing countries and has been authorized in 2020 within the European Union (EU) according to its novel food regulation. This article reviews current knowledge on the safety of coffee leaf tea. From the various ingredients contained in coffee leaves, only two were highlighted as possibly hazardous to human health, namely, caffeine and epigallocatechin gallate (EGCG), with maximum limits implemented in EU legislation, which is why this article focuses on these two substances. While the caffeine content is comparable to that of roasted coffee beans and subject to strong fluctuations in relation to the age of the leaves, climate, coffee species, and variety, a maximum of 1–3 cups per day may be recommended. The EGCG content is typically absent or below the intake of 800 mg/day classified as hepatotoxic by the European Food Safety Authority (EFSA), so this compound is suggested as toxicologically uncritical. Depending on selection and processing (age of the leaves, drying, fermentation, roasting, etc.), coffee leaf tea may exhibit a wide variety of flavors, and its full potential is currently almost unexplored. As a coffee by-product, it is certainly interesting to increase the income of coffee farmers. Our review has shown that coffee leaf tea is not assumed to exhibit risks for the consumer, apart from the well-known risk of caffeine inherent to all coffee-related beverages. This conclusion is corroborated by the history of its safe use in several countries around the world.

## 1. Introduction

Coffee consumption has been known worldwide for decades. As early as 1450, pilgrims brought coffee from Ethiopia to Mecca and during the 17th century to Europe [1,2]. However, the so-called coffee by-products, such as coffee flowers, leaves, pulp, husk, parchment, and silver skin, are rarely known outside of the countries of manufacture [3,4].

The *Coffea* genus consists of several species and approximately 6000 varieties [5]. The most important coffee species are *C. arabica* and *C. canephora*.

*C. arabica* accounts for around 60–75% of global production and grows in plantations at high altitudes of 600–2000 m, whereas *C. canephora* accounts for the remaining proportion and grows in plantations from 300–800 m [5,6]. For completeness, the species *C. liberica* and *C. excelsa* are still cultivated to a limited extent [5,7].

*Coffea* spp. needs a consistently warm and at the same time humid climate without extremes. Therefore, the plants grow in tropical and subtropical regions between the 25th latitude north and south along the equator [5], the so-called “coffee belt” [8,9]. The main producing countries include Brazil, Mexico, Guatemala, Colombia, Honduras, Vietnam, Indonesia, India, Ethiopia, and Uganda [5].

As mentioned by Haller et al. [10], *Camellia sinensis* tea is the most widely consumed drink in the world directly after water. It is obvious that making tea from coffee leaves may offer economic potential in the coffee-growing regions [11]. Figure 1 and Figure 2 show examples of coffee leaves on coffee plants.

According to Rimbach et al. [5], around half of German consumers drink tea every day. That means that every year around 30 L of tea (250 g of tea leaves/year) is consumed, although there are large regional differences. In contrast, coffee consumption is higher. Finland, for example, has the highest coffee consumption per capita. Every resident drinks an average of five cups of coffee a day, while in Germany, the average consumption is four cups a day or 6.7 kg of coffee beans per year [5]. According to the International Coffee Organization, exports of all forms of coffee amounted to almost 119 million bags (60 kg bags) for the coffee year 2020/21 (October 2020–August 2021) [12].

Food and beverages can be marketed on the European Union (EU) market without authorization if there was a significant human consumption within the EU before 15 May 1997 (see EU’s guidance on “human consumption to a significant degree” [13]). If this was not the case, a particular food can still be authorized according to the novel food regulation, which also provides a separate pathway for notification of traditional foods from third countries [14]. As described by Chen et al. [15], coffee leaf tea is a well-known traditional beverage in coffee plant-growing countries, for example, “copi daon” in Indonesia, “giser” in Yemen or “Kahwa dau” in West Sumatra, Indonesia [15,16]. From the authors’ own travel experience, we can say that coffee leaf tea is less known among tourists and still rarely offered. For illustration, Figure 3 shows some examples of brewed coffee leaf tea, whereas Figure 4 shows some examples of dried and chopped coffee leaves for tea infusions. Samples (Figure 3 and Figure 4) were prepared using large tasting bowls (Figure 5) according to International Organization for Standardization (ISO) standard ISO 3103 [17].

The company AM Breweries (Denmark) submitted a notification request [18] to the EU commission according to Article 14 “Notification of a traditional food from a third country” (EU regulation 2015/2283) [14]. Before granting authorization, the European Commission can ask the European Food Safety Authority (EFSA) for a scientific risk assessment [19]. In 2020, the EFSA published its ‘Technical Report on the notification of infusion from coffee leaves… as a traditional food from a third country’ [20]. Subsequently, an implementing regulation was published [21] that allows coffee leaf tea to be placed on the EU market (positive list in the annex [21,22]) as a traditional food from a third country, in accordance with the novel food regulation (regulation 2015/2283) [14]. Since the applicant AM Breweries [18] did not provide evidence of the use of coffee leaves as an ingredient in other beverages, only the infusion of coffee leaves was approved as a novel food [21,22]. Among other factors, such as microbiological criteria, critical values have been established for the substances caffeine and (−)-epigallocatechin-3-gallate (EGCG) [22].

## 2. Materials and Methods

For this article, electronic searches of the literature were conducted, including the databases PubMed (National Library of Medicine, Bethesda, MD, USA) and Google Scholar (Google LLC, Mountain View, CA, USA). A wide range of search terms was used, including coffee, coffee leaf tea, caffeine, and EGCG. In addition, standard works of literature were used concerning knowledge of the product. Furthermore, databases such as the US Department of Agriculture (USDA) database for the flavonoid content of selected foods were searched for terms such as caffeine and EGCG, as well as the EFSA homepage for the terms coffee leaf tea, caffeine, and EGCG.

In addition to the data from the literature, our own data of the *C. arabica* varieties (Bourbon and Pacamara) of El Salvador from a recent Master’s thesis were considered [23]. The 14502-1 test regulations that were used to determine the total polyphenol content in tea and ISO 14502-2, used to determine the catechin content in green tea, were used [24].

## 3. Composition of Coffee Leaf Tea

### 3.1. Ingredients

Caffeine and EGCG are naturally found in coffee leaves [15,25]. As described by Zheng et al. [26], caffeine is a purine alkaloid found mainly in young leaves and shoots including buds and cotyledons, whereas caffeine is not detected in roots, aged cotyledons or older brown parts of shoots [26]. The results of the aforementioned publication suggest that caffeine accumulation is specific to the above-ground parts (leaves, cotyledons, and shoots) of the seedlings and that biosynthesis is performed in young tissues. The authors describe caffeine synthesis as a defense mechanism of the (soft) parts of plants against predators [26]. Frischknecht et al. [27] found that the formation of purine alkaloids is costly for the plant [27].

Plant-based phenols such as EGCG are products of the plant’s secondary metabolism [6]. In contrast to black tea, green tea is neither fermented nor oxidized. Due to that, polyphenols such as epicatechin (EC), epicatechin-3-gallate (ECG), epigallocatechin (EGC), and epigallocatechin gallate (EGCG) can make up about 59% of the catechin content [10]. The proportion of phenolic compounds in fresh, young tea leaves of the tea plant is high [5]. Rimbach et al. [5] mention that four-fifths of these compounds are made up of catechins (flavan-3-ols) and gallocatechins (flavan-3,4-diols), which belong to flavanols [5]. According to Frede et al. [6], fresh tea shoots contain several flavanols (catechins), for example, epigallocatechin gallate (9–13%), epicatechin gallate (3–6%), and epigallocatechin (3–6%), as well as epicatechin (1–3%) and others (catechin, gallocatechin, 1–2%) [6].

The infusion of coffee leaves is a beverage rich in several bioactive compounds (e.g., chlorogenic acids, xanthones [28], trigonelline, adenine-7-glucosyl, quercetin, and mangiferin [29,30], not only polyphenols (catechins) and caffeine described above.

Basically, the EFSA has assessed the ingredients and focused on caffeine and EGCG. All other compounds did not pose any significant risk. This article therefore focuses on these two compounds, knowing that there are other ingredients in coffee leaf tea.

### 3.2. Analytical Results

In a recent project conducted with coffee leaves from El Salvador, various coffee leaf teas were analyzed, among other factors, for their caffeine and EGCG content [23]. The samples examined differed in terms of the age of the leaves used, drying, fermentation, and other processing conditions (see examples in Table 1). A total of 24 different samples were produced.

According to the results of the study (Table 2), the samples contained 3.4 g/kg DW (dry weight) to 23.9 g/kg DW of caffeine, and catechins from not detectable to 6.5 g/kg DW (sum of all analyzed catechins, only epicatechin and epicatechin gallate were present) [23,24]. EGCG was not detected in the samples [23,24]. The moisture content of the samples ranged from 3.9 to 16.1% [23,24]. Figure 6 shows some dried coffee leaves.

Ratanamarno et al. [25] found that the amount of these substances depends on the age of the leaf [25]. They recorded higher amounts in young leaves than in full-grown leaves. Fibrinato et al. [31] came to the same conclusion [31]. Acidri et al. [32] showed that total phytochemicals, for example, catechins, in coffee leaves decreased with leaf age [32]. However, Steger [23] could not confirm this for the dried coffee leaf teas examined, as no significant correlation with the age of the coffee leaves was observed (Figure 7).

#### 3.2.1. Catechins

Samples 1–24 *C. arabica*: The results of the total catechin content show a clear influence of the blending process (Figure 8). Except for one sample, all blended samples have no catechins at all (Figure 7). According to scientific research this could be due to the oxidation of catechins by polyphenol oxidase to theaflavins or similar oligomeric structures [33]. This effect also occurs during black tea fermentation [34]. The large surface area and added water of the mixed samples could be responsible for an enhanced enzymatic reaction. Furthermore, the drying parameters show an influence on the total catechin content. Air drying has a significantly higher average value (0.266 g/100 g DW) than oven drying (0.054 g/100 g DW) and a significantly lower value than roasting (0.479 g/100 g DW) [23] (Figure 8).

Krol et al. [35] report that the polyphenol content decreases as the coffee beans are roasted. This is because polyphenolic components are thermolabile and easily disintegrate at temperatures above 80 °C. This process occurs due to the thermal instability during roasting [35]. Our tests on coffee leaves could not confirm this. The highest catechin contents were measured here in the roasted samples [23]. Further research will have to show whether there is a difference between coffee beans and coffee leaves in this regard.

Li et al. [36] investigated the correlation between the temperature and duration of the thermal treatment with the catechin content. In accordance with this, the low content of the oven-dried sample can be attributed to the 4-h drying time. The roasted samples had a much shorter drying time (20 min), resulting in the highest content (Figure 8). The EGCG mentioned in the European Commission’s approval [22] could not be detected in any sample. Turkmen et al. [37] stated that the absence of EGCG in black tea was due to the oxidation and fermentation processes. This effect could have also occurred in the coffee leaf tea samples during the withering process.

Samples 25–28, *C. canephora*: Samples of *C. canephora* do not contain catechins, according to Figure 7. No data on this result could be found during scientific research. Our four samples represent only a small spot check. Therefore, this result should be verified by examining a larger number of samples.

#### 3.2.2. Caffeine

Samples 1–24 *C. arabica*: The caffeine content of the leaves varies between 0.37 g/100 g of DW and 1.33 g/100 g DW. Here, the type of leaf and variety show significant influences on the caffeine content of the tea. Young leaves show the highest caffeine value (0.91 g/100 g DW) while yellow leaves show the lowest (0.44 g/100 g DW). Caffeine levels for *C. arabica* are approximately the same as those detected by Ratanamarno et al. [25].

The effect of caffeine reduction with leaf age has already been observed in some studies with different plants, e.g., *Ilex paraguariensis* and *Plantago lanceolata* [38,39]. Song et al. [40] explained this effect mainly by the function of caffeine as a pesticide. Younger leaves of the plant must be more protected compared to older ones; therefore, the plant builds higher concentrations in these leaves. Furthermore, processing is a significant variable toward caffeine content. Here, rolling (1.2 g/100 g DW) and freezing (1.00 g/100 g DW) of the leaf show the highest caffeine content. According to Astill et al. [41] the caffeine content decreases during the fermentation and drying stage. In the case of freezing, it is possible that metabolic pathways are stopped, resulting in less degradation during drying. Since the rolling process was conducted on young leaves, further tests would be needed to determine whether this influences the caffeine content. Furthermore, the low caffeine content of the blended samples may be due to the addition of water. Some of the caffeine may have been dissolved in the water during processing and then dripped off through the drying bed during drying. The caffeine content in the infusion prescribed in the approval of the EU Commission [22] can be exceeded depending on the dose of the coffee leaves per water.

Samples 25–28, *C. canephora*: The *C. canephora* samples have the highest levels of caffeine according to Figure 9. Rodriges et al. [42] and Perrois et al. [43] also showed that *C. canephora* has a higher caffeine content than *C. arabica*. Miedaner confirmed this observation [44].

Figure 10 shows the caffeine content depending on the leaf type, variety, and processing based on our samples (*C. arabica*) from El Salvador (coffee plantation: Finca la Quintanilla and Finca la Palma).

The higher caffeine content of *C. canephora* coffee leaves (Figure 8) is plausible in correspondence with the higher caffeine content in *C. canephora* beans [44,45].

## 4. Exposure Assessment

### 4.1. National Consumption Study

The National Consumption Study II (NVS II, Nationale Verzehrsstudie II) [46,47] of the Max Rubner Institute (MRI) served as a data basis concerning the consumption of tea infusions in adolescents and adults (Table 3). NVS II is still the most up-to-date representative study on the food and beverage consumption of the German population. In this study, around 20,000 people between the ages of 14 and 80 were asked about their consumption habits. It was conducted across Germany between 2005/2006.

This study includes consumption information for individual days (24-h method, [46]). Therefore, this study is suitable for exposure estimates for both acute and chronic risks. Individual body weights of the participants were used as a basis for intake estimates [46]. Table 3 shows the average results for men and women. For simplicity, details on the individual age groups are not given here but can be found in the study.

The daily value for fluid intake is at least 1.5 L/day according to NVS [46]. The amount of nonalcoholic beverages consumed hardly differs between men (2.351 g/day) and women (2.285 g/day) [47]. About one-third of this amount is covered by coffee and (black/green) tea (men: 571 g/day or 24%; women: 506 g/day or 22%), while the consumption of herbal tea/fruit tea is much lower. The proportion of herbal tea and fruit tea is more than twice as high in women (318 g/day or 14%) than in men (149 g/day or 6%) [47].

For further considerations, the total value for coffee and tea is used since coffee leaf tea may replace either coffee or tea in the daily diet in a worst-case scenario.

The term “tea” refers to infusing plant parts in hot water. However, according to the definition in the German Food Book, the term tea may be used only for products made from the tea plant (*Camellia sinensis*). Other products made from leaves, fruits or flowers (e.g., mint, rosehip, hibiscus) are grouped under the term tea-like products (sometimes also referred to as herbal tea or tisane). Therefore, coffee leaf tea is under legal consideration a tea-like product [48]. Nevertheless, the exposure of coffee leaf tea should better be assessed using the data from the caffeine-containing coffee/tea category than the caffeine-free herbal tea category.

In the approval of coffee leaf tea under the novel food regulation, maximum levels have been established for some substances that must be met to sell the product [22]. These include caffeine, chlorogenic acid (5-CQA), and EGCG. Table 4 lists the limit values according to the Novel Food Regulation [22]. Two of the substances (caffeine and EGCG) are discussed in more detail below.

### 4.2. Caffeine Exposure

The taste of Arabica coffee is particularly aromatic, soft, and mild. In contrast, *C. canephora* coffee of most commercial qualities is perceived to be less delicate and develops a higher caffeine content [44]. In addition to many ingredients such as flavorings and aromas, the coffee beans of both varieties contain caffeine. In small amounts, this primarily affects the central nervous system and increases the capacity for mental absorption and memory while at the same time the resulting fatigue is reduced [44]. Therefore, it is not surprising that coffee has an image as a stimulant (central nervous system stimulant) [44,45]. However, consuming it in excessive quantities can cause anxiety, insomnia as well as nausea, vomiting, diarrhea, and gastrointestinal upset [44,45]. Table 5 lists some examples of beverages concerning the caffeine content. Average values of caffeine in selected foods are shown in Table 6.

The oral LD_50_ of caffeine in rats is 192 mg/kg, and the iv LD_50_ in rats is 105 mg/kg [45]. An acute fatal overdose of caffeine in humans is approximately 10–14 g (equivalent to 150–200 mg/kg of body weight) [45].

An estimated 90% of adults in the United States consume caffeine daily, which means that caffeine intake is almost universal. The average daily intake is 200 mg caffeine/day, which according to Table 5 corresponds to approximately two cups of filter coffee. There are no known effects on the liver. Epidemiological studies suggest that regular coffee consumption has a modest protective effect against the progression of chronic liver disease and the development of liver cancer [45,50].

High levels of caffeine were observed in energy drinks, in particular, see Table 5 [49]. Several reports of liver damage related to the use of caffeine-rich energy drinks have been published, but they are, as described in the source, uninformative as they are not well documented, and the causality is questionable [45]. Therefore, caffeine is unlikely to cause liver damage, but the various high-caffeinated energy drinks widely available can possibly cause liver damage if used in excess [45]. The likelihood score for caffeine in coffee is rated E, which means that it is an unlikely cause of clinically apparent liver injury [45].

According to the WHO assessment of the carcinogenicity of caffeine in humans and also in laboratory animals, there is insufficient evidence that caffeine is carcinogenic. Caffeine was therefore classified as group 3, i.e., not classifiable, concerning its carcinogenicity for humans [51]. Intoxication with caffeine is also included in the World Health Organization’s International Classification of Diseases (ICD-10) [45]. Caffeine levels of 57 mg/kg intravenously or oral doses of 18–50 g are lethal in human adults [45].

Zhang et al. [52] showed the preventive benefits of coffee and tea or their combination with respect to stroke and dementia. Two to three cups of coffee/day or three to five cups of tea/day or their combined consumption of four to six cups/day was shown to have the lowest hazard ratio (HR) for stroke and dementia.

As mentioned above, *C. arabica* coffees normally have a lower caffeine content (0.9 and 1.4 g/100 g DW) compared to *C. canephora* varieties (1.5 and 2.6 g/100 g DW) [5,6]. According to Rimbach [5], the caffeine content in tea is 1–4% of the dry weight on average.

Analysis of coffee leaf tea made of *C. arabica* var. Pacamara and Bourbon showed amounts between 0.34 and 1.24 g/100 g or 0.4–1.3% of the dry weight [23,24] (Table 7).

Here, the type and variety of leaves had significant influences on the caffeine content of the tea (Figure 10). Young leaves had the highest caffeine value (0.91 g/100 g DW) while yellow leaves had the lowest (0.44 g/100 g DW) [23].

The *C. canephora* samples showed amounts between 1.91 and 2.39 g/100 g or 2.1–2.6% of the dry weight. Fibrianto et al. reported that “coffee leaves contain caffeine at 21.9 g/kg of dry weight” [31]. There is 87.6 mg caffeine/cup of coffee leaf tea. This roughly corresponds to the content of coffee (Table 6). As shown in Table 2, the caffeine content of coffee leaf tea has a wide range since it is a natural raw material.

The EFSA published a scientific opinion on the safety of caffeine [53] in 2015. As mentioned by EFSA “caffeine intakes of no concern derived for acute caffeine consumption by adults (3 mg/kg bw per day) may serve as a basis to derive single doses of caffeine and daily caffeine intakes of no concern for these population subgroups”. According to the Ministry of Food and Agriculture (BMEL), Germany, and the EFSA, up to 200 mg of caffeine as a single dose or 400 mg spread over a day is supposed to be virtually free of risk for healthy adults [49,53]. Table 7 lists some examples of beverages concerning the caffeine content and dosage.

**Table 7 foods-11-00263-t007:** Caffeine Content.

Product	Portion Size	Dosage per Drink	Amount of Liquid [mL]	Caffeine Content [%/DW]
**Coffee**				
Coffee *(C. canephora*, roasted) [54]	1 cup	4 g	125–150	1.5–3.0
Coffee (*C. arabica*, roasted) [54]	1 cup	4 g	125–150	1.2–1.5
**Tea**				
Black tea [54]	1 cup	2 tea spoons	300	1.3–4.0
Green tea [54]	1 cup	2–3 heaped tea spoons	300	0.1–5.9
Coffee leaf tea *(C. arabica)*	1 cup	5 g	250 (1 tea cup)	0.4–1.3
Coffee leaf tea *(C. canephora)*				2.1–2.6
Rooibos-tea [54]	No caffeine			

For tea infusions, it is mentioned that significantly less caffeine dissolves in water depending on the preparation [54]. According to Frede et al. [6], the transfer of substances into the drink depends on the size of the leaf and brewing time and the water temperature. For a 2-min infusion (tea bag), Frede et al. [6] mentioned that nearly 100% of caffeine and around 50% of catechins pass into the drink while other sources [54] mention that not all caffeine dissolves in the water during the preparation of the infusion.

The caffeine content is not only influenced by the *Coffea* spp. used but also by brewing temperature and time. The recommended optimal brewing conditions as reported by Hariyadi et al. [55] are shown in Table 8. As already described above, the maximum caffeine content is <80 mg/L [22]. To not exceed this value, several scenarios were calculated in Table 9.

The maximum daily caffeine intake possible with coffee leaf tea consumption depends on its content in the plant. With *C. canephora,* the recommended amounts are much lower than with *C. arabica.* The *C. canephora* samples (Samples 25–28) have the highest levels of caffeine, according to Figure 8. Rodrigues et al. [42] and Perrois et al. [43] also showed that *C. canephora* has a higher caffeine content than *C. arabica*. Miedaner confirmed this observation [44]. The calculations were performed as an example of the minimum and maximum caffeine content of the measured coffee leaf teas measured.

*C. arabica*: With the minimum caffeine content, up to five cups (200 mL) can be consumed, while with the maximum caffeine content, only one cup would be advisable.

*C. canephora*: With the minimum caffeine content, one cup (200 mL) can be consumed, while with the maximum caffeine content, half a cup would be advisable.

The mathematical values represent the worst-case considerations. Today, three to five cups of coffee a day are considered safe for healthy people [44]. Naturally, the assessment also depends on the individual dosage, with higher dosages such as 80 g coffee leaves per L of water often being preferable for taste reasons.

Information on the number of coffee/tea cups also fits well with the protective effects concerning stroke and dementia [52]. The caffeine content depends on the type of plant used and the processing of the coffee leaves. Therefore, the daily intake considered safe in terms of the caffeine content is between half a cup and five cups of coffee leaf tea (Table 9).

Since caffeine levels correspond largely to those of coffee, but the limit value of caffeine in coffee leaf tea is comparably low [22], the national consumption study (NVS) [46,47] should also be placed into focus. According to the NVS, the intake of herbal teas is not high. Values with health risk due to caffeine are not reached with normal tea consumption.

The price of coffee leaf tea is currently high compared to other tea products, so it is probably not consumed daily.

### 4.3. Exposure to (−)-Epigallocatechin-3-Gallate (EGCG)

According to Haller et al. [10], polyphenols are the most common antioxidants. Naturally, they are available in fruits, vegetables, grains, legumes, chocolate, and beverages such as tea, coffee or wine [10]. According to the USDA database, EGCG is found in many everyday foods, such as dairy products, apples or bananas [56].

In 2018, EFSA was asked to provide a scientific opinion on green tea catechins [57]. Evaluation of the ingredient EGCG, as the main representative of catechins, can also be applied to the tea made from coffee leaves. The EFSA panel considered the possible association between the consumption of EGCG, the most relevant catechin in green tea, and hepatotoxicity [57]. According to EFSA, EGCG is present in unconjugated form in plasma after oral ingestion and it is the most cytotoxic catechin (compared to EGC and ECG) in primary rat hepatocytes. In this assessment, EFSA concluded that the amount of catechins absorbed through *Camellia sinensis* tea infusions is safe. However, it should be noted that on rare occasions, liver damage has occurred after ingesting green tea (more than 800 mg of EGCG/day). The reason given is likely an idiosyncratic reaction [57]. The panel states that ‘hepatotoxicity is the adverse effect listed in the compendium, which also states that the doses causing hepatotoxicity are not indicated in human case reports’ [57]. Additionally, there are also positive effects of EGCG, such as reduced risk of type 2 diabetes and its cardiovascular complications [58], prevention of metabolic syndrome [59], improved endothelial function and insulin sensitivity, reduced blood pressure, and protection against myocardial injury [60]. The combination of EGCG (blood pressure lowering) and caffeine (stimulating, blood pressure-increasing effect) in coffee leaf tea could be an interesting area of research.

The daily intake of tea is shown in Table 3. According to the table, men consume 538 g/day and women 506 g/day of green tea on average every day. The quantities of herbal tea are much lower (Table 3). According to EFSA, an adult consumes an average of 90–300 mg of EGCG/day. When consumed in large amounts, the value can reach 866 mg [57]. Similar to coffee leaf tea, the chemical composition of green tea varies depending on the type of plant, environment, season, age of the leaves, and thus also the EGCG content, which is in the range of 1600 to 20,320 mg/100 g of dried leaves (13 times). Regarding green tea infusions, the EGCG content varied over more than an 88-fold range (2.3–203 mg/100 g of infusion) [57].

By evaluating studies, the EFSA concluded that there is no evidence of hepatotoxicity below 800 mg of EGCG/day for up to 12 months [57]. EFSA concludes that catechins from (green) tea infusions and similar beverages are generally safe.

According to Figure 7, catechins could only be detected in *C. arabica*, i.e., not in *C. canephora* coffee leaf tea [23,24]. Since the *C. canephora* leaf teas examined were only a small sample, further research is necessary.

The EGCG content, which is potentially liver damaging (above 800 mg of EGCG/day), does not relate to tea drinks but to extracts [57]. Therefore, from a toxicological perspective, the caffeine content is the focus of attention. However, a value of more than 800 mg of EGCG/day is not reached with normal daily coffee leaf tea consumption.

## 5. Conclusions

In contrast to black and green tea infusions, which are consumed as traditional foods, coffee leaf tea contains higher levels of caffeine, with the content roughly corresponding to that of coffee beans. The amount of caffeine in coffee leaf tea depends on the processing, the plant (species, variety), the harvest season, the age of the leaves, and the beverage preparation (dosage, temperature, etc.). Due to the caffeine content, up to three cups of coffee leaf tea, drunk throughout the day, are considered safe. Since coffee leaf tea is a niche item in Europe, it is unlikely that this amount will be exceeded.

The positive effects of caffeine (see Section 4.2) and EGCG (Section 4.3) are less obvious due to the (still) low consumption of coffee leaf tea and should be examined more closely. No EGCG could be detected in our coffee leaf tea samples examined. The results of the analytical data (all species and varieties) for caffeine and EGCG are below the EFSA Safety Thresholds and can be considered safe with moderate consumption in terms of toxicity or side effects.

According to Miedaner [44], it takes around nine months before coffee fruits can be harvested. In the press, it has been repeatedly reported that the current coffee prices are well below the production costs. Since it is a plant with evergreen leaves, its leaves can be harvested all year long and therefore represent an additional source of income for coffee farmers, as coffee beans and leaves are from the same plant [11].

Finally, it should be noted that EGCG and caffeine in coffee leaf tea occur in a specific matrix rich in polyphenols. This results in differences regarding bioavailability and effects compared with the isolated substances investigated in most studies.

## Figures and Tables

**Figure 1 foods-11-00263-f001:**
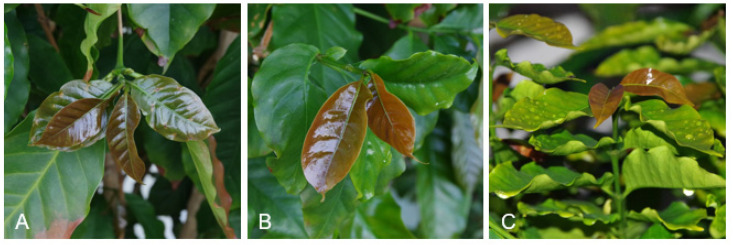
Coffee leaves (*C. arabica*) ((**A**–**C**): strong, dark brown, new shoots).

**Figure 2 foods-11-00263-f002:**
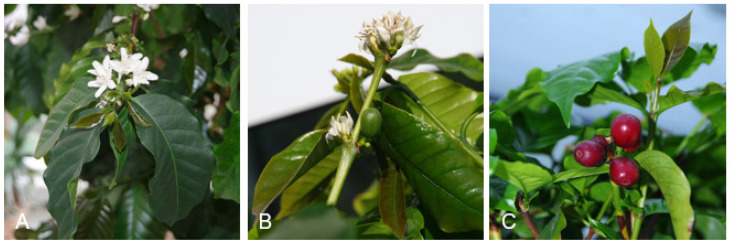
Fresh shoots on branches with flowers (**A**,**B**) and fruits (**C**) (*C. arabica;* newly sprouted leaves are darker than older ones).

**Figure 3 foods-11-00263-f003:**
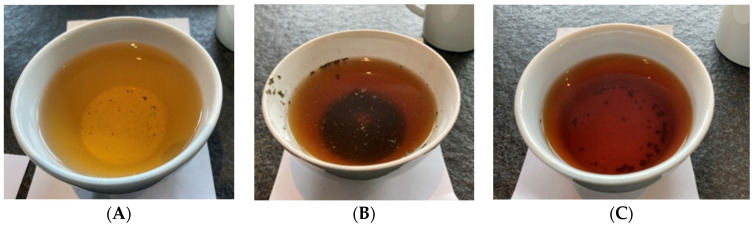
Examples of brewed coffee leaf tea infusions (water extraction)—different processing shows different colors. Preparation: 2 g of dried coffee leaves/100 mL water ((**A**) = Bourbon, old leaves, air/sun dried, wild fermentation; (**B**) = Pacamara, old leaves, whole leaves, oven-dried, no fermentation; (**C**) = Bourbon, old leaves, whole leaves, roasted, no fermentation).

**Figure 4 foods-11-00263-f004:**
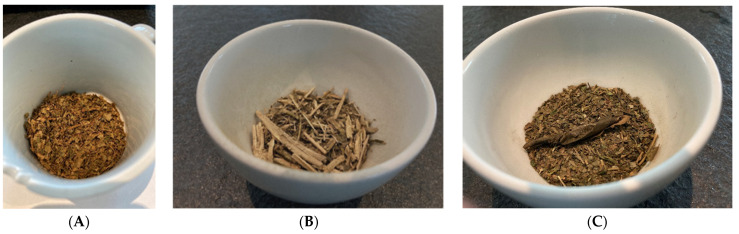
Examples of dried coffee leaf tea ((**A**) = Pacamara, yellow leaves, mixed, air/sun dried, fermentation by yeast; (**B**) = Bourbon, whole water shoots, chopped, 1-h steam distilled, air/sun dried; (**C**) = Bourbon, old leaves, mixed, air/sun dried, wild fermentation).

**Figure 5 foods-11-00263-f005:**
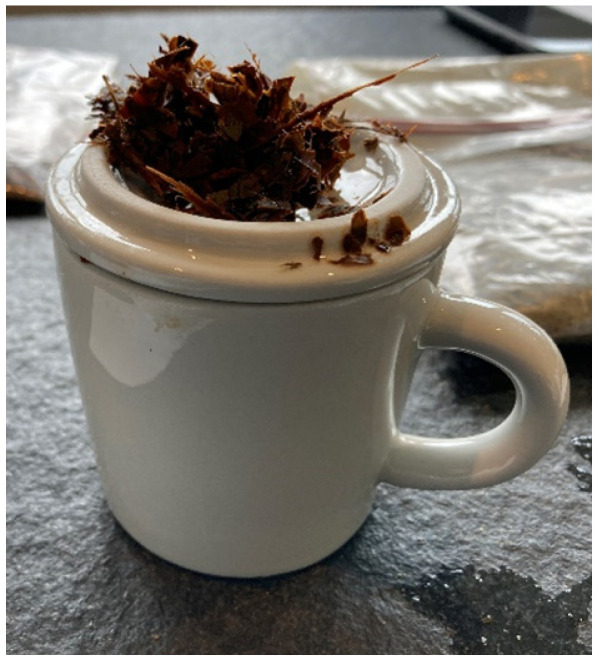
Spent coffee leaves after the preparation of a coffee leaf tea infusion.

**Figure 6 foods-11-00263-f006:**
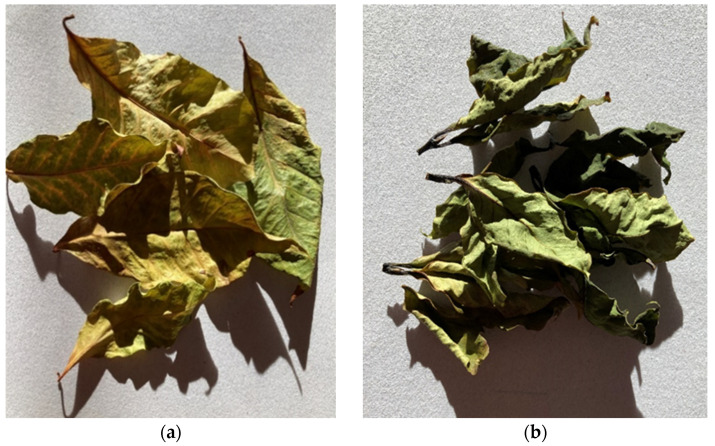
Air-dried coffee leaves, *C. arabica*. (**a**) Coffee leaves that had fallen from the bush themselves, old leaves. (**b**) Coffee leaves, fresh shoots, only brown leaves. (**c**) Coffee leaves, fresh shoots, only the green leaves. (**d**) Coffee leaves, fresh shoots, mixture of leaves from branches/shoots.

**Figure 7 foods-11-00263-f007:**
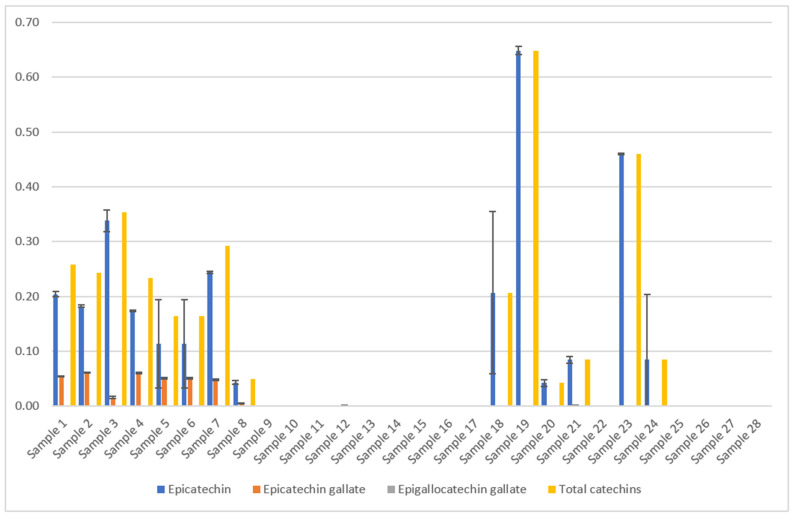
Results of the coffee leaf tea samples examined for the catechin content [23,24], analyzed by HPLC. The mean value in g/100 g DW is given from three determinations in each case. The total catechin content depends on the drying method and processing. Epigallocatechin gallate could not be detected in the samples examined. The error bars show the standard deviation of the three measured values. The total catechin contents were calculated as the sum of the mean values of all catechins (only epicatechin and epicatechin gallate were detected). Samples 1–24 picked in El Salvador from two different plantations, *C. arabica* [23]. Samples 25–28 are Indian *C. canephora*. [Sample 1: Pacamara old whole air Palma, Sample 2: Pacamara yellow whole air Palma, Sample 3: Pacamara old crumbled air Palma, Sample 4: Pacamara yellow crumbled air Palma, Sample 5: Pacamara old cutted air Palma, Sample 6: Pacamara old blended air Palma, Sample 7: Pacamara old crumbled air Palma Yeast fermented, Sample 8: Pacamara old whole oven Palma, Sample 9: Pacamara old blended air Palma Lacto., Sample 10: Pacamara old blended air Palma Yeast, Sample 11: Pacamara old blended air Palma Wild, Sample 12: Pacamara yellow blended air Palma Wild, Sample 13: Pacamara yellow blended air Palma Yeast, Sample 14: Pacamara yellow blended air Palma Lacto., Sample 15: Bourbon water shoot whole blended air Quintanilla wild, Sample 16: Bourbon water shoot/old blended air Quintanilla Yeast, Sample 17: Bourbon water shoot/old blended air Quintanilla Wild, Sample 18: Bourbon water shoot/young green tea air Quintanilla, Sample 19: Bourbon water shoot/young black tea air Quintanilla Wild, Sample 20: Bourbon water shoot/whole blended/steamed air Quintanilla, Sample 21: Bourbon water shoot/old whole air Quintanilla wild, Sample 22: Bourbon water shoot/old whole/frozen air Quintanilla, Sample 23: Water shoot old whole roasted Quintanilla, Sample 24: Water shoot whole air Quintanilla, Sample 25: India prepackage, Sample 26: India bulkware, Sample 27: India CxR, Sample 28: India old Paradenia. Yeast = *Saccharomyces cerevisiae, var. bayanus*, lacto. = *Lactobacillus plantarum*].

**Figure 8 foods-11-00263-f008:**
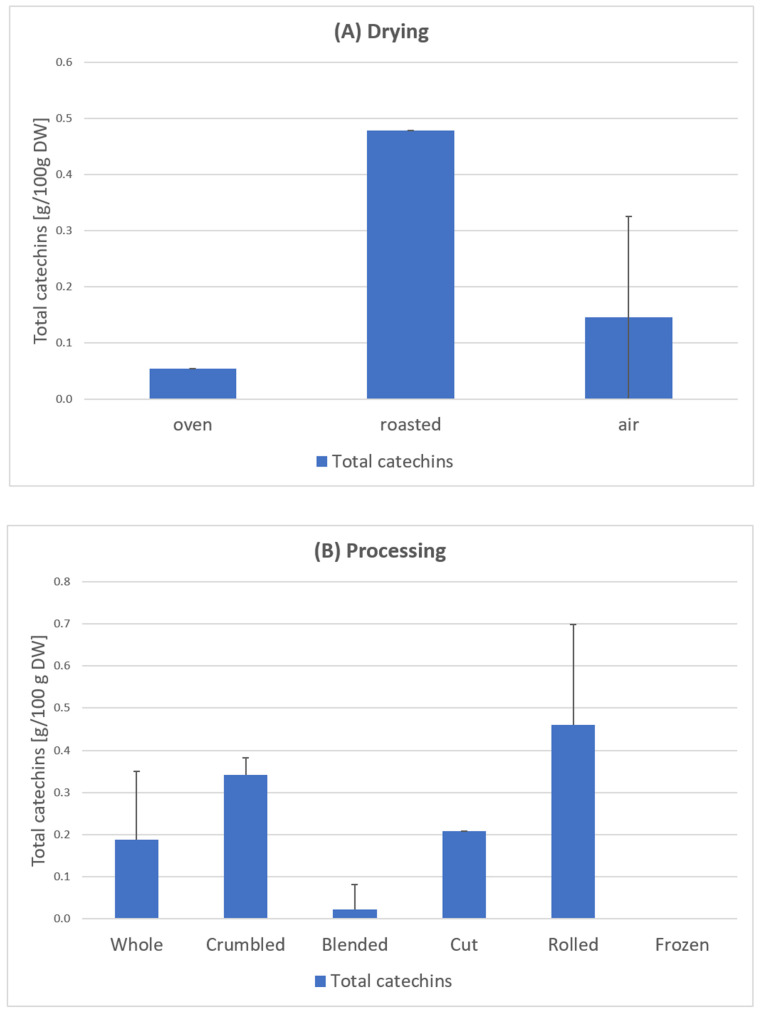
Total catechin content in leaves depending on the drying method (**A**) and processing (**B**), based on samples of *C. arabica* from El Salvador [23]. Description of the samples, see Figure 7. Drying: Oven-dried sample 8; roasted samples 23; air-dried samples 1–7; 9–22, 24. Processing: Whole samples 1–2, 8, 21–24, Crumbled samples 3–4, 7; Blended samples 6, 9–17, 20; Cut sample 5; Rolled sample 19; Frozen sample 22. The results show considerable variances depending on the individual processing and drying steps (sample 1–24).

**Figure 9 foods-11-00263-f009:**
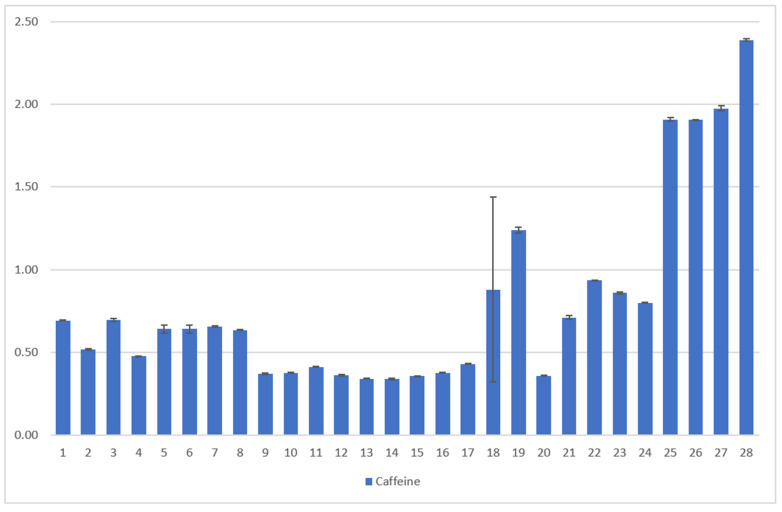
Results of the coffee leaf tea samples examined for the caffeine content [23,24], analyzed by HPLC. The mean value in g/100 g DW is given from three determinations in each case. The error bars show the standard deviation of the three measured values. The first 24 samples were collected in El Salvador from two different plantations and were dried and fermented in different ways, *C. arabica* [23]. The last four samples are Indian *C. canephora*. [Description of the samples, see Figure 7].

**Figure 10 foods-11-00263-f010:**
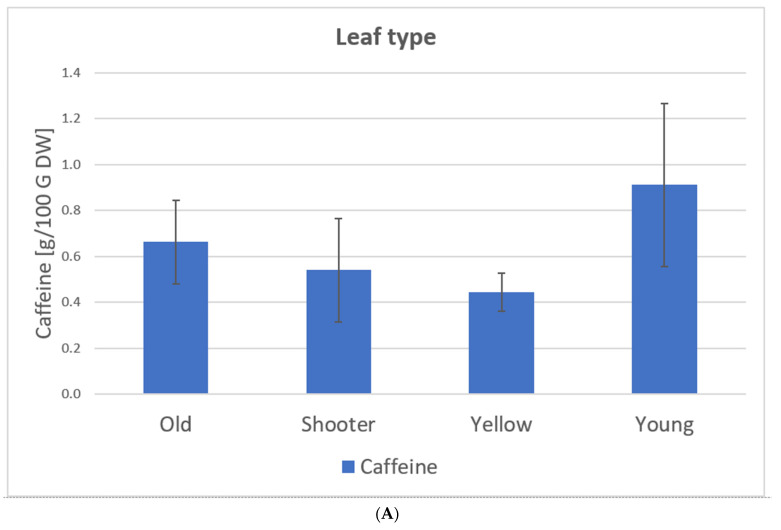
Caffeine content depending on leaf type (**A**), variety (**B**), and processing (**C**) based on the samples from El Salvador [23], *C. arabica*. Description of the samples, see Figure 7. (**A**): Old samples 1, 3, 5–11, 16, 21–23; Water shoot samples 15, 20, 24; Yellow samples 2, 4, 12–14; 17–19. (**B**): Pacamara samples 1–14; Bourbon samples 15–24. (**C**): Whole samples 1, 2, 8, 21, 23–24: Crumbled samples 3, 4, 7; Blended samples 6, 9–17, 20; Cut sample 5; Rolled samples 18–19; Frozen samples 22.

**Table 1 foods-11-00263-t001:** Some examples of coffee leaf tea samples prepared in El Salvador.

Picture	Processing		
	*Coffea arabica* Variety	Leaves	Drying, Fermentation
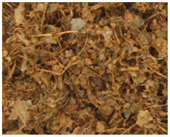	Pacamara	old leaves, mixed	air/sun dried, fermentation by yeast (*Saccharomyces cerevisiae,* var. *bayanus*)
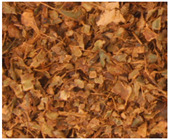	Pacamara	yellow leaves, mixed	air/sun dried, fermentation by yeast (*Saccharomyces cerevisiae,* var. *bayanus*)
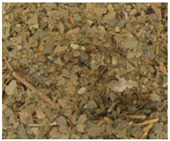	Pacamara	old leaves, whole leaves	Oven-dried, no fermentation
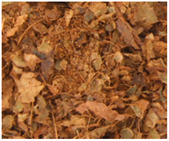	Pacamara	yellow leaves, mixed	air/sun dried, wild fermentation
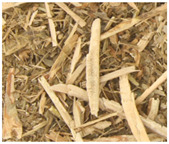	Bourbon	whole water shoots, chopped	1 h steam distilled, air/sun dried
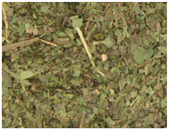	Bourbon	old leaves, whole leaves	air/sun dried, no fermentation
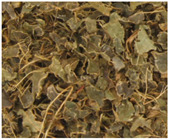	Bourbon	old leaves, mixed	air/sun dried, wild fermentation
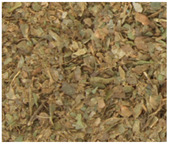	Bourbon	old leaves, whole leaves	roasted, no fermentation

**Table 2 foods-11-00263-t002:** Analytical results for fresh and dried coffee leaves.

	Fresh Coffee Leaves (mg/g) [25]	Dried Coffee Leaves (mg/g DW) [23,24]
Caffeine	1.8–3.2	3.4–23.9
Epigallocatechin gallate	5.5–16.4	n.d.
Epicatechin gallate	0.26–0.48	n.d.–0.6
Epicatechin	0.27–0.40	n.d.–6.5
Catechins, sum	0.05–0.18	n.d.–6.5

n.d., not detectable.

**Table 3 foods-11-00263-t003:** NVS (National Consumption Study) results [46].

	Average Male (*n* = 6160)		Average Female (*n* = 7593)	
[g/day]	Arithmetic Average	Confidence Interval	Arithmetic Average	Confidence Interval
Coffee and tea (black/green)	538	527/550	506	497/515
Herbal tea/fruit tea	133	125/141	251	241/261

**Table 4 foods-11-00263-t004:** Maximum permitted levels of caffeine, 5-CQA, and EGCG as approved by the European Commission [22].

Substance	Maximum Amount
Caffeine	<80 mg/L
5-CQA	<100 mg/L
EGCG	<700 mg/L

**Table 5 foods-11-00263-t005:** Average values for selected foods [49].

Beverage	Caffeine Amounts ^a^
Filter coffee	90 mg/200 mL
Energy drink	80 mg/250 mL
Black tea	45 mg/200 mL
Coke	35 mg/330 mL
Dark chocolate	25 mg/50 g
Iced-tea	17 mg/330 mL

^a^ It should be noted that these are plant-based, natural raw materials and the caffeine content can therefore vary [49].

**Table 6 foods-11-00263-t006:** Caffeine amount/cup [25].

Beverage	Caffeine Amounts
Coffee ^a^	74 mg/237 mL
Tea (*Camellia sinensis*)	15 mg/237 mL
Coffee leaf tea ^a^	12 mg/250 mL

^a^ Without specifying the variety, probably *C. arabica*, as according to Miedaner, around two-thirds of the coffee sold worldwide is made from Arabica beans [44].

**Table 8 foods-11-00263-t008:** Recommended optimal processing point [55].

	*Canephora* Coffee Leaf Tea	*Liberica* Coffee Leaf Tea
Brewing conditions: temperature [°C]	93.4	91.7
Brewing conditions: duration [min]	4.80	4.84
caffeine level [mg/100 mL]	74.9	72.5

**Table 9 foods-11-00263-t009:** Maximum daily caffeine intake based on coffee leaf tea.

Product	Caffeine Content [g/100 g] [23,24]	Virtually Safe Intake (Coffee Leaf Tea) ^a^
Coffee leaf tea *(C. arabica)*	Min. 0.34 Max. 1.24	Min. caffeine content: 1.18 L Max. caffeine content: 0.32 L
Coffee leaf tea *(C. canephora)*	Min. 1.91 Max. 2.39	Min. caffeine content: 0.21 L Max. caffeine content: 0.17 L

^a^ The following assumptions were made: Complete (100%) extraction of caffeine. Weight of water [g] = volume of water [mL]. To prepare the coffee leaf tea, 2 g of dried coffee leaves/100 mL of infusion is required (which corresponds to typical manufacturer’s information). Calculation: Caffeine content [g/100 g]/50 = caffeine content in 2 g of dried coffee leaves [1]. The caffeine content given in g/100 g of tea, calculated at the 2 g of tea used [1]. Caffeine content in 2 g of dried coffee leaves with a maximum caffeine content of 0.8 g/100 g [2]. How often the caffeine content of 2 g of coffee leaf tea falls within the safe value of 0.8 g of caffeine [2]. Finally, the value is recalculated from 100 mL to 1 L [3].

## Data Availability

No new data were created or analyzed in this study. Data sharing is not applicable to this article.

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
