# Peer review of "Risk Assessment of Caffeine and Epigallocatechin Gallate in Coffee Leaf Tea"

_foods, 2022, doi:10.3390/foods11030263_

Round 1

Reviewer 1 Report

  1. the authors should summarize the toxicity or side-effects of caffeine and EGCG by referring to the reported results.
  2. only caffeine and EGCG were discussed in 3.1 Ingredients. Other components and their safety should be discussed.
  3. the authors use “Figures 1 and 2” in Line 55 and “Figs 3 and 4” in Line 81.
  4. check the word Epicatechingallat in Figure 7.

Author Response

1. the authors should summarize the toxicity or side-effects of caffeine and EGCG by referring to the reported results.

Some further details on toxicity and side-effects of caffeine and EGCG were added to the paper as requested.

2. only caffeine and EGCG were discussed in 3.1 Ingredients. Other components and their safety should be discussed.

Basically, the EFSA has assessed the ingredients and focused on caffeine and EGCG. All other compounds did not pose any risk. We therefore focused our review on these two compounds. A detailed review of all other components would go beyond the scope and length of this article.

3. the authors use “Figures 1 and 2” in Line 55 and “Figs 3 and 4” in Line 81.

The in-text references to Figures were corrected. Thank you for identifying this issue.

4. check the word Epicatechingallat in Figure 7.

The word was corrected in Figure 7.

Reviewer 2 Report

The authors intended to evaluate the safety of drinking coffee leaf tea just by reviewing the existing literature and considering substances that can be only toxic if ingested in very large amount. It is well known that coffee leaf tea has been traditionally consumed in different cultures and its consumption in the West in very low. So this is not new. Furthermore, this reviewer does not think that only by evaluating the content of caffeine catechins and total polyphenols one can affirm that coffee leaf tea is safe. The title leads the reader to thinking that the manuscript will introduce new compounds that could be toxic to humans, which is not the case. Perhaps, the approach of the paper could change and other compounds could also be included.

Also, the authors conclude that coffee leaf tea is safe but they don´t mention the differences among the types of leaf tea evaluated.

In some parts of the manuscript, the content seems to be confusing, with a large variety of information, some of them unnecessary, like the graphs containing the intake of liquid foods by men and women (the main information could be only in the text). Therefore, the manuscript would benefit from reorganization.

Author Response

The authors intended to evaluate the safety of drinking coffee leaf tea just by reviewing the existing literature and considering substances that can be only toxic if ingested in very large amount. It is well known that coffee leaf tea has been traditionally consumed in different cultures and its consumption in the West in very low. So this is not new. Furthermore, this reviewer does not think that only by evaluating the content of caffeine catechins and total polyphenols one can affirm that coffee leaf tea is safe. The title leads the reader to thinking that the manuscript will introduce new compounds that could be toxic to humans, which is not the case. Perhaps, the approach of the paper could change and other compounds could also be included.

The literature on coffee leaf tea is rather limited, and we were unable to identify compounds other than the ones already known that may pose a risk. As remarked above, the text is already very long, so that we currently fail to see the need to add anything else. The title was changed to better represent the content of the paper, and to avoid expectations about other compounds we are currently not able to fulfil.

Also, the authors conclude that coffee leaf tea is safe but they don´t mention the differences among the types of leaf tea evaluated.

The species and varieties are now discussed in more detail in section 5.

In some parts of the manuscript, the content seems to be confusing, with a large variety of information, some of them unnecessary, like the graphs containing the intake of liquid foods by men and women (the main information could be only in the text). Therefore, the manuscript would benefit from reorganization.

As requested, we have deleted the figure about liquid intake, and tried to refocus the whole paper on the essential information during the revision.

Reviewer 3 Report

Dear Authors,

The work raises an interesting guess.
it is badly organized. I believe that traditional literature searches would give better results. The reference in the manuscript to "from a recent master thesis were considered" is unfortunate.
The manuscript should be redrafted and referred to the most recent data available. Currently, 70% is literature from the last 10 years and there are too few scientific articles.
The manuscript must be compiled in accordance with the editorial requirements.

Author Response

The work raises an interesting guess.
it is badly organized. I believe that traditional literature searches would give better results. The reference in the manuscript to "from a recent master thesis were considered" is unfortunate.

Actually, as stated in the methods section, we have conducted traditional literature searches in several databases. In addition to that, we have used data from a recent master thesis conducted at our institutes by one of the co-authors (M.S.). We do not see anything inappropriate about this approach.

The manuscript should be redrafted and referred to the most recent data available. Currently, 70% is literature from the last 10 years and there are too few scientific articles.

As the topic of coffee leaf tea and all other coffee by-products is only of recent interest, most papers were published within the last 10 years. There is not much, if any, evidence before that. The traditional knowledge about the beverage was often published in non-science articles, but we rather cite these than nothing.

Please also refer to our recent assessment of coffee by-product literature [1]: “Before 2010, only a few papers were published about the topic starting with a rather visionary early report on the applications of coffee by-products published in 1938 by do Amaral, where several of the fields mentioned in this conference were already suggested. Subsequently, only a few articles on the use of coffee pulp as animal feed were published in the 1970s and 1980s.”

As a note, none of these references from before the 1990s were about coffee leaf tea.

References:

[1] Lachenmeier, D.W.; Schwarz, S.; Rieke-Zapp, J.; Cantergiani, E.; Rawel, H.; Martín-Cabrejas, M.A.; Martuscelli, M.; Gottstein, V.; Angeloni, S. Coffee By-Products as Sustainable Novel Foods: Report of the 2nd International Electronic Conference on Foods—“Future Foods and Food Technologies for a Sustainable World”. Foods 2022, 11, 3. https://doi.org/10.3390/foods11010003

The manuscript must be compiled in accordance with the editorial requirements.

We have strictly followed the information for authors as well as the journal template.

Reviewer 4 Report

The Review of the manuscript “Is There a Risk Associated with the Traditional Consumption of Coffee Leaf Tea?”

The Authors of the manuscript review the information about coffee leaves tea and risk exposure of caffeine and catechins on consumers’ health. The infusion of coffee leaves is a beverage rich in a lot of bioactive compounds not only polyphenols (catechins and chlorogenic acids and caffeine described in the review. There is a lot of information in the recent articles about the content of many chemical compounds determined in coffee leaves and coffee leaf tea with the use of HPLC as well as LC-MS/MS. So in my opinion there is a lack of important information in this review. There is a lack of information about such compounds in coffee leaves as mangiferin and isomangiferin. Trigonelline content in coffee leaves is almost in the same amount as caffeine. The level of chlorogenic acids in fresh leaves could be on the similar level as in green coffee beans. In my opinion the information about these compounds and amount of them in coffee leaves and coffee leaf tea should be added.

Below are some missing articles:

Chen XM, Ma Z, Kitts DD. Effects of processing method and age of leaves on phytochemical profiles and bioactivity of coffee leaves. Food Chem. 2018; 249: 143-153.

Chen X, Mu K, Kitts DD. Characterization of phytochemical mixtures with inflammatory modulation potential from coffee leaves processed by green and black tea processing methods. Food Chem. 2019; 271: 248–258.

Monteiro A, Colomban S, Azinheira H, Guerra-Guimarães L, Do Céu Silva M, Navarini L, Resmini M. Dietary antioxidants in coffee leaves: impact of botanical origin and maturity on chlorogenic acids and xanthones. Antioxidants. 2020; 9: E6.

In my opinion these article should be added and some information about coffee beans should be removed – Table 3.

Detailed points to correct:

l. 78 leave– should be leaves

l. 158 leave - the same as above

l. 287-288 why there is a lack of information about chlorogenic acids in coffee leaves?

l. 387 should be “80 mg/L” not “80 g/L”

All figures should be corrected (apart from photos), especially Figure 7. This figure is totally unclear. Lines with SD for the results are not correct for samples 9-17, 22, 25-28. It should be changed and corrected.

Figure 8 – SD lines should be corrected

There is a lack of Figure 12.

Author Response

The Authors of the manuscript review the information about coffee leaves tea and risk exposure of caffeine and catechins on consumers’ health. The infusion of coffee leaves is a beverage rich in a lot of bioactive compounds not only polyphenols (catechins and chlorogenic acids and caffeine described in the review. There is a lot of information in the recent articles about the content of many chemical compounds determined in coffee leaves and coffee leaf tea with the use of HPLC as well as LC-MS/MS. So in my opinion there is a lack of important information in this review. There is a lack of information about such compounds in coffee leaves as mangiferin and isomangiferin. Trigonelline content in coffee leaves is almost in the same amount as caffeine. The level of chlorogenic acids in fresh leaves could be on the similar level as in green coffee beans. In my opinion the information about these compounds and amount of them in coffee leaves and coffee leaf tea should be added.

Our scope was the components with adverse, but not the bioactive compounds in general. As remarked in response to the other reviewers, we have clarified the scope of the paper by change in title. Nevertheless, thank you for the references below. We have included them for completeness.

Below are some missing articles:

Chen XM, Ma Z, Kitts DD. Effects of processing method and age of leaves on phytochemical profiles and bioactivity of coffee leaves. Food Chem. 2018; 249: 143-153.

Chen X, Mu K, Kitts DD. Characterization of phytochemical mixtures with inflammatory modulation potential from coffee leaves processed by green and black tea processing methods. Food Chem. 2019; 271: 248–258.

Monteiro A, Colomban S, Azinheira H, Guerra-Guimarães L, Do Céu Silva M, Navarini L, Resmini M. Dietary antioxidants in coffee leaves: impact of botanical origin and maturity on chlorogenic acids and xanthones. Antioxidants. 2020; 9: E6.

In my opinion these article should be added and some information about coffee beans should be removed – Table 3.

The articles were added as requested and Table 3 was removed. Thank you for providing these references.

Detailed points to correct:

  1. 78 leave– should be leaves

The spelling of leaf/leaves was corrected throughout.

  1. 158 leave - the same as above

The spelling of leaf/leaves was corrected throughout.

  1. 287-288 why there is a lack of information about chlorogenic acids in coffee leaves?

There is no "lack". The 5-CQA was just not the focus of our publication. We had selected caffeine and EGCG. The EU also specifies a maximum content for 5-CQA, this is quoted in the table.

  1. 387 should be “80 mg/L” not “80 g/L”

No, we actually mean g/L here (this is the dosage of the tea leaves in water). We have clarified the sentence to make this clearer.

All figures should be corrected (apart from photos), especially Figure 7. This figure is totally unclear. Lines with SD for the results are not correct for samples 9-17, 22, 25-28. It should be changed and corrected.

Fig. 7+9 have been revised and the error bars for measured values with "0" have been removed. An explanation of the calculation has also been included. In sample 18, the large error bars are consistent with the measured values. We have re-checked this again using the raw data. All figures were also revised for better visual clarity.

Figure 8 – SD lines should be corrected

We have added explanations under the figure which of our samples were used for the calculation to make it clearer. The error bars are partly quite large due to natural variations and influences of processing.

There is a lack of Figure 12.

This was a typo. Figure 11 was referenced. Due to the deletions and renumbering, we have carefully re-checked all intext references to the tables and figures.

Round 2

Reviewer 2 Report

The authors did not make all the sugested changes but it is ok for me the way it is.

Author Response

Thank you for your re-assessment of our paper!

Reviewer 4 Report

All corrections were made. Only Figure 8 total catechins should not below 0.

Author Response

Thank you for your carful reading of our paper and your further comment.

Figure 8 has been corrected as advised.